# Explainable Transfer Learning on Graphs Using a Novel Label Frequency Representation

## Abstract

Graphs are characterized by their versatility in representing objects from a wide range of domains, such as social networks or protein structures. This flexibility and power poses a significant challenge for transfer learning between graph domains. Current methods of transfer learning between graph domains tend to focus exclusively on the structure of the underlying graphs, neglecting the characteristics of the nodes and not addressing the difficulties in comparing nodes that represent very dissimilar entities, such as atoms and people for instance. In this paper, we propose a novel universal representation of graphs based on the relative frequency of the node labels. This novel representation enables explainable transfer learning between labeled graphs from different domains for the first time, without the need for additional adaptations. That is, we show that our novel representation can be readily combined with a data alignment technique that in turn allows transfer learning between data from different domains. Experimental results show that knowledge can be acquired from graphs belonging to chemical and biological domains to improve the accuracy of classification models in social network analysis. A comparison with state-of-the-art techniques indicates that our approach outperforms existing non-topological methods and, in some cases, even graph neural networks. In summary, our technique represents a major advance in graph node representation for transfer learning between different domains, opening up new perspectives for future research.

## 1 Introduction and Related Work

In pattern recognition and machine learning, graphs are often used for representing complex data. The reason for this is twofold. First, graphs can explicitly model the intricate relationships that might exist between entities, and second, graphs can adapt their size and complexity to the size and complexity of the actual data objects. For instance, graphs can be used to describe the structure of molecules (Hu et al., 2020a), where nodes can be used to represent atoms and edges might encode the chemical bonds between the atoms. Another example can be found in social network analysis (Yanardag & Vishwanathan, 2015), where nodes might represent persons and edges then indicate their relationships. Graphs are also successfully applied in image representations (Li & Gupta, 2018), where nodes can represent clusters of pixels, for instance, and edges can then be used to measure the similarity between these clusters.

Despite the growing interest in graph-based pattern recognition and machine learning, both their complex structures and the scarcity of suitable datasets (Manchanda et al., 2023) pose significant research challenges. These challenges somehow limit the application of diverse powerful pattern recognition and machine learning techniques to graphs. In recent years, solutions have been proposed to address these obstacles, such as the expansion of datasets by data augmentation techniques (Fuchs & Riesen, 2022), or the use of unsupervised approaches for pre-training networks (Hu et al., 2020b). Another promising approach would be to adapt transfer learning to the graph domain. Transfer learning is a machine learning technique in which a model, pre-trained to perform a task, is reused and optimized to face a related (but different) task by exploiting previously acquired knowledge (Pan & Yang, 2010).

Some approaches for transfer learning on graphs have been proposed. For example, (Zhao et al., 2023) and (Roncoli et al., 2023) focus on transfer learning between graphs that share similar seman-

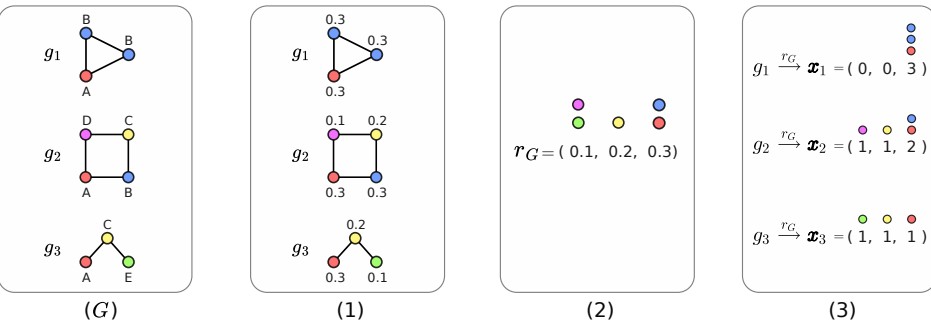

Figure 1: Illustration of creating the novel representation for three sample graphs $g_1$, $g_2$ and $g_3$ stemming from a dataset $G$ based on the label frequencies.

tics, viz. social networks and physical environments, respectively. However, generalization of these methods to completely different domains remains problematic. Some studies (Zhu et al., 2021) have explored joint learning techniques that attempt to overcome these limitations. These approaches have shown promising potential, but their ability to generalize across very different domains is still limited. Other research directions (Verma & Zhang, 2019) propose dataset-specific solutions that require domain-specific adaptations.

In this paper, we propose a novel representation of graphs that facilities the transfer of learning between graphs of very different domains. In particular, our approach is based on a frequency analysis of the underlying node labels including a frequency alignment operation between different graph datasets. By representing graphs and their nodes as frequency vectors, the method becomes applicable to any dataset (regardless of what the nodes represent). That is, unlike existing solutions, our representation does not require specific adaptations. Moreover, as suggested by recent proposals (Gillioz & Riesen, 2023), we focus on nodes and their labeling rather than the global topology. This is also corroborated by recent studies suggesting that relatively simple non-topological methods for graph classification can compete with more sophisticated techniques. For example, it has been shown that Global Sum Pooling, followed by a Multilayer Perceptron (MLP), can achieve similar, or better, results than more complex models on some datasets (Errica et al., 2020). Last but not least, our novel method not only achieves general applicability and high flexibility but also a high degree of explainability, thus enabling a clear understanding of how and why transfer learning occurs.

The remaining paper is structured as follows. In Section 2, we detail the creation and operation of the representation itself as well as the transfer learning process. In Section 3, we present an experimental evaluation using two chemical datasets to improve the classification performance on a social network dataset. Finally, in Section 4, we draw conclusions, discuss the limitations of our approach and propose possible future research directions.

## 2 METHODS

In this section, we first introduce our novel representation of graphs based on relative node label frequencies (described in Section 2.1), and then discuss the alignment of these representations in a common space (explained in Section 2.2).

### 2.1 REPRESENTING GRAPHS AS ABSOLUTE FREQUENCY VECTORS

Graphs are mathematical structures useful for modeling complex entities and their relationships. Formally, a graph can be defined as a triple $g = (V, E, \mu)$, where $V$ is the set of nodes, $E \subseteq \{(u, v) | u, v \in V, u \neq v\}$ is the set of edges, and $\mu : V \to L$ is a function that assigns a label $\mu(v) = l \in L$ to each node $v \in V$ (where $L$ is a given label alphabet). Sometimes this definition is complemented with a second function, which also assigns a label to the edges in the graph – in this paper, however, we only use node-labeled graphs (in fact, the methods proposed in this section could also be adapted to labeled edges).

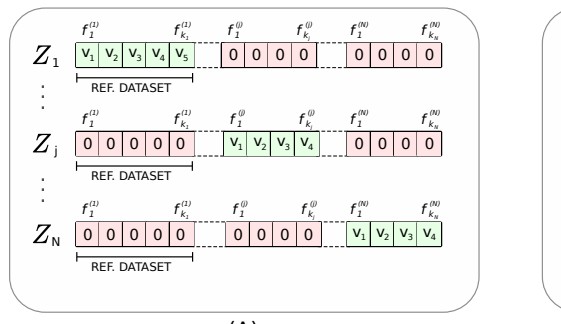 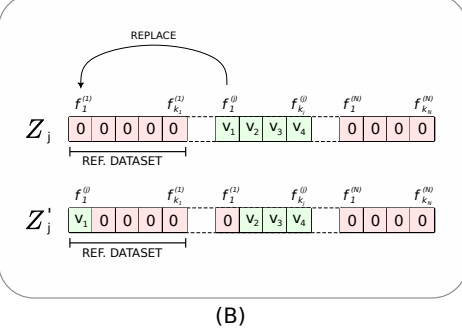

(A)                    (B)

Figure 2: Illustration of the alignment process between relative frequency vectors.

From now on, we suppose that the label alphabet $L$ consists of $n$ discrete symbols $l_1, \ldots, l_n$, and the dataset of graphs $G = \{g_1, \ldots, g_N\}$ consists of $N$ graphs of the type $g = (V, E, \mu)$. To carry out the transfer of knowledge, we propose to create a new representation of the graphs consisting of a vector of relative frequencies of all labels $l_i \in L$. In a first step, for each label $\mu(v) = l_i \in L$ that can be observed in the dataset $G$, we compute its relative frequency $f(l_i)$ with respect to all $N$ graphs from $G$ and round its value to the fourth decimal place (see Fig. 1, Step 1). Next, we define a $k$-dimensional vector termed $\boldsymbol{r}_G = (f_1, \ldots, f_k)$, which is common to all graphs in the same dataset $G$ (see Fig. 1, Step 2). This vector $\boldsymbol{r}_G$ contains all the relative frequencies $f_i$, that actually occur on the nodes of all graphs from $G$, in ascending order (without duplicates). Since multiple labels in $L$ may have the same relative frequency, the size $k$ of $\boldsymbol{r}_G$ may be smaller than the total number $n$ of unique labels in $L$. Next, for each graph $g_j \in G$ we count how often each relative frequency $f(l_i) \in \boldsymbol{r}_G$ appears among the nodes of $g_j$ based on the vector $\boldsymbol{r}_G$. That is, we replace $g_j \in G$ with a $k$-dimensional vector $\boldsymbol{x}_{G_j} = (\nu_1, \ldots, \nu_K) \in \mathbb{R}^n$ where $\nu_i$ is the absolute frequency of the relative frequency $f_i$ actually observed on the nodes in set $V_j$ of $g_j$. This produces a representation that reflects the distribution of relative label frequencies into a vector-based representation of the graph (see Fig. 1, Step 3).

To ensure that only information from the training set is actually used in this transformation (and thus avoid data leakage), the dataset must be split into a training and a test set before calculating the new representation. Of course, this can lead to a situation where we see a particular label, for which we need to calculate the relative frequency, for the first time in the test set. In such cases, we use the minimum relative frequency for this unknown label (present in $\boldsymbol{r}_G$) to represent the corresponding label. Since the label was not observed in the training set (and therefore tends to have a low relative frequency), this heuristic seems reasonable.

## 2.2 ALIGNMENT OF REPRESENTATIONS FOR TRANSFER OF KNOWLEDGE

In the previous section, we show how we can transform graphs from arbitrary graph datasets into a new $k$-dimensional vector representation. In the following, we assume that we have $N$ different graph datasets $G_1, \ldots, G_N$ available. For each of these $N$ datasets, we calculate the relative frequency vectors $\boldsymbol{r}_{G_1}, \ldots, \boldsymbol{r}_{G_N}$ of size $k_1, \ldots, k_N$ respectively. By means of this information, we now have a $k_i$-dimensional vector space for each of the $N$ graph datasets, in which the graphs are mapped ($i = 1, \ldots, N$). Note that, each vector space (or the associated graph domain) can have different dimensions and represent objects from very different domains.

The transfer of knowledge between these different domains is now achieved by aligning the $N$ separate graph representations in a common representation space. In this paper, our main goal is to preserve explainability during the transfer process. Thus, instead of using methods for learning the representations (Ganin et al., 2016), which might improve the transfer of knowledge but neglect interpretability, we thus propose the following suboptimal alignment approach.

The proposed alignment strategy (described in Algorithm 1) requires a machine learning model $M$ (e.g., a classification model) and a metric $\beta$, that measures the model performance (e.g., the classification accuracy of $M$). In addition to these two inputs, the algorithm expects $N$ common spaces

---

**Algorithm 1** Dataset Alignment $(M, \beta, Z_1, \ldots, Z_N)$

1: Train Model $M$ on $Z_1$
2: **for** $1, \ldots, K$ **do**
3:     **for** each $j \in \{2, \ldots, N\}$ **do**
4:         evaluate $Z_j$ with model $M \to \beta_\Theta$
5:         **for** each index $(j, p) \in \{(j, 1), \ldots, (j, k_j)\}$ **do**
6:             **for** each index $(1, t) \in \{(1, 1), \ldots, (1, k_1)\}$ **do**
7:                 $Z'_j = Z_j$
8:                 replace features with indices $(j, p)$ and $(1, t)$ in $Z'_j$
9:                 evaluate $Z'_j$ with model $M \to \beta_j$
10:               **if** $\beta_j > \beta_\Theta$ **then**
11:                   $\beta_\Theta = \beta_j$
12:                   $Z_j = Z'_j$
13:               **end if**
14:             **end for**
15:         **end for**
16:     **end for**
17: **end for**

---

$Z_1, \ldots, Z_N$ (for each relative frequency vector $\boldsymbol{r}_{G_1}, \ldots, \boldsymbol{r}_{G_N}$). The dimension of all these common spaces $Z_i$ is equal to $\sum_{j=0}^{N} k_j$, i.e., the sum of the sizes of the $N$ frequency vectors $r_{G_1}, \ldots, r_{G_N}$.

The common space based on the relative frequencies $\boldsymbol{r}_{G_1}$ associated with the graph dataset $G_1$ is termed $Z_1$ from now on. In vector $Z_1$, we have actual values between indices $1, \ldots, k_1$, while the remaining values at indices $(k_1 + 1), \ldots, N$, are set to 0. In general, we consider the representation $Z_j$, associated to the relative frequency vector $\boldsymbol{r}_{G_j}$, with values uniquely between indices $k_{j-1} + 1$ to $k_j$, while all other values are set to 0 (see Fig. 2 (A)). For clarity of the indices, we introduce the following notation. Each entry in vector $Z_j$ can be referred to with an index pair $(q, p)$, where $q$ represents one of the $N$ dataset graphs and lies between $1, \ldots, N$, and $p$ represents one of the $k_j$ frequencies of the relative frequency vector $\boldsymbol{r}_{G_j}$ and lies between $1, \ldots, k_j$. The valid indices of any vector space $Z_j$ are therefore $((1, 1), \ldots, (1, k_1), (2, 1), \ldots, (2, k_2), \ldots, (N, k_n))$.

Algorithm 1 starts by training the model $M$ on the training set $Z_1$ to distinguish classes based on the first indices $(1, 1), \ldots, (1, k_1)$ (line 1 of Algorithm 1). After that, in a loop (starting at line 3), for each dataset $G_j$ with $j = 2, \ldots, N$, the corresponding representation $Z_j$ is evaluated by model $M$ resulting in a metric $\beta_\Theta$. The value $\beta_\Theta$ is the basis for the next comparisons, since the examined dataset has not yet been aligned and therefore has no values in the first indices $(1, 1), \ldots, (1, k_1)$. Now, modified versions $Z'_j$ of $Z_j$ are constructed in two further for loops (starting at line 5 and 6 respectively). Formally, during these loops, replacement of the elements with indices $(j, p)$ and $(1, t)$ are applied (with $p$ between $1, \ldots, k_j$, and $t$ between $1, \ldots, k_1$) (see Fig. 2 (B)). This new representation $Z'_j$ is then again analyzed by model $M$ (line 9), yielding a new value $\beta_j$. If $\beta_j$ is greater than the reference value $\beta_\Theta$, the change is consolidated by updating both the reference value $\beta_\Theta$ and the representation $Z_j$ that led to this improvement.

The procedure of Algorithm 1 can be repeated in a global loop up to $K$ times (starting at line 2), making the overall computational complexity of the algorithm equal to $O(K \times N \times \max\limits_{j \in \{2, \ldots, N\}} (k_j) \times k_1)$. This iterative approach allows the alignment to be progressively refined, optimizing the dataset representations with respect to the reference dataset.

## 3 EXPERIMENTAL EVALUATION AND DISCUSSION

The experimental evaluation consists of four parts. In Section 3.1, we provide a visual analysis that allows us to observe the transfer of knowledge between significantly different domains. Next, in Section 3.2, we investigate the impact of the transfer of knowledge in a classification experiment. In Section 3.3, a comparison of our best model with state-of-the-art methods is presented. Finally, in Section 3.4, we conduct an ablation study to verify the benefit of having similar distributions.

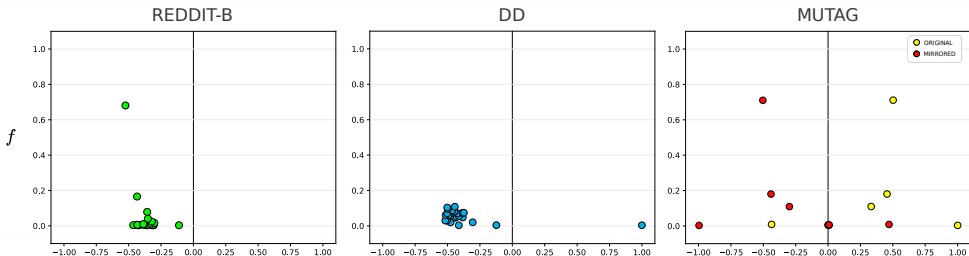

Figure 3: Distributions of node label frequencies $f$ (on the Y-axis) with respect to class memberships (on the X-axis). The X-axis represents the normalized imbalance of the labels with respect to one of the two classes (values close to -1 for negative classes, values close to +1 for positive classes).

All experiments are conducted on three publicly available graph datasets (Morris et al., 2020) viz. DD and MUTAG, which contain graphs representing proteins and molecules, respectively, and REDDIT-BINARY (REDDIT-B), that contains graphs representing social networks. For DD and MUTAG we use the original labels on the nodes, for REDDIT-B we use the node degree as labels (the nodes of these graphs are unlabeled). Further details on the datasets can be found in Appendix A.3.

### 3.1 VISUAL ANALYSIS

The differences in the number of nodes with the same relative frequency for each class of graphs are crucial for both classification (Bishop & Nasrabadi, 2007) and transfer learning. To visualize these differences, we present the plots shown in Fig. 3 for all datasets. Each dot in Fig. 3 represents a unique relative frequency $f$ on the Y-axis and the corresponding relative class memberships on the X-axis. That is, on the Y-axis dots close to 1 represent common labels, dots close to 0 indicate uncommon labels. On the X-axis we show the normalized difference in the relative number of nodes associated with the same relative frequency from the graph classes. Values close to -1 indicate that more nodes of graphs from the negative class have this frequency value $f$, while values close to +1 indicate that more nodes of graphs from the positive class have this frequency value $f$ (an X-value of 0.0 means that half of the nodes having frequency $f$, originate from one or the other class).

When we compare the distributions of the three datasets, we notice the following. First, the rather rare labels of REDDIT-B are similar to the rare labels of DD in terms of their relative class membership. Our basic hypothesis is that these similarities can be used to transfer the knowledge between these two domains. If, on the other hand, one compares REDDIT-B with MUTAG, there is no obvious correspondence. Therefore, for the MUTAG dataset, we also show the distribution of frequencies when the underlying classes are mirrored – this dataset will be referred to as MUTAG-M from now on. We now see that the common labels of REDDIT-B are similarly distributed to those of MUTAG-M. Hence, the presence of many nodes with a common label should increase the probability of a graph being labeled negatively, regardless of the actual domain.

To summarize, we see that the distribution profiles of REDDIT-B, DD and MUTAG-M are visibly similar despite the different domains, which at least makes a transfer of knowledge possible.

### 3.2 EVALUATION OF TRANSFER OF KNOWLEDGE

For the following evaluations, we use four state-of-the-art classification models, viz. a K-Nearest Neighbors ($k$-NN), a Support Vector Machine (SVM), a Random Forest (RFC) and a Multi-Layer Perceptron (MLP). Each model is trained with class balancing and optimized by cross-validated grid search (exclusively applied on the reference dataset). Details of the evaluation parameters for all models are given in Appendix A.2. To evaluate the performance, we use a *stratified K-Fold cross-validation* with *K=10*, measuring the average balanced accuracy in a binary classification scenario. This metric ensures a baseline accuracy of 50% for random classification, which is essential for testing the discrimination ability of our novel representation.

|  | | TEST DATASET | | |
|---|---|---|---|---|
| | | **DD** | **MUTAG-M** | **REDDIT-B** |
| **REF. DATASET** | **DD** | $78.0 \pm 1.6$ | $68.6 \pm 9.9$ | $76.6 \pm 4.5$ |
| | **MUTAG-M** | $71.6 \pm 3.2$ | $82.0 \pm 5.3$ | $67.8 \pm 1.9$ |
| | **REDDIT-B** | $73.1 \pm 3.8$ | $81.1 \pm 7.7$ | $79.3 \pm 2.1$ |

Table 1: $k$-NN model

|  | | TEST DATASETS | | |
|---|---|---|---|---|
| | | **DD** | **MUTAG-M** | **REDDIT-B** |
| **REF. DATASET** | **DD** | $77.5 \pm 2.3$ | $50.0 \pm 0.0$ | $50.0 \pm 0.0$ |
| | **MUTAG-M** | $70.3 \pm 6.0$ | $81.6 \pm 5.8$ | $67.0 \pm 3.9$ |
| | **REDDIT-B** | $68.1 \pm 7.7$ | $80.9 \pm 7.9$ | $83.9 \pm 1.9$ |

Table 2: RFC model

|  | | TEST DATASETS | | |
|---|---|---|---|---|
| | | **DD** | **MUTAG-M** | **REDDIT-B** |
| **REF. DATASET** | **DD** | $77.8 \pm 3.9$ | $51.5 \pm 4.6$ | $57.7 \pm 6.1$ |
| | **MUTAG-M** | $59.6 \pm 8.4$ | $83.4 \pm 5.1$ | $65.8 \pm 5.0$ |
| | **REDDIT-B** | $58.8 \pm 6.9$ | $55.9 \pm 9.7$ | $80.3 \pm 1.6$ |

Table 3: MLP model

|  | | TEST DATASETS | | |
|---|---|---|---|---|
| | | **DD** | **MUTAG-M** | **REDDIT-B** |
| **REF. DATASET** | **DD** | $80.1 \pm 3.4$ | $50.0 \pm 0.0$ | $50.0 \pm 0.0$ |
| | **MUTAG-M** | $50.0 \pm 0.0$ | $82.3 \pm 6.7$ | $51.6 \pm 3.4$ |
| | **REDDIT-B** | $77.9 \pm 4.0$ | $78.8 \pm 9.7$ | $81.1 \pm 2.6$ |

Table 4: SVM model

| | $k$-NN | | RFC | | MLP | | SVM | |
|---|---|---|---|---|---|---|---|---|
| **DATASETS** | **ORIGINAL** | **JOINT** | **ORIGINAL** | **JOINT** | **ORIGINAL** | **JOINT** | **ORIGINAL** | **JOINT** |
| **DD** | $78.0 \pm 1.6$ | $\mathbf{78.9 \pm 2.1}$ | $77.5 \pm 2.3$ | $\mathbf{78.7 \pm 2.7}$ | $77.8 \pm 3.9$ | $\mathbf{78.0 \pm 4.4}$ | $80.1 \pm 3.4$ | $\mathbf{80.2 \pm 3.3}$ |
| **MUTAG-M** | $82.0 \pm 5.3$ | $\mathbf{82.7 \pm 5.3}$ | $81.6 \pm 5.8$ | $\mathbf{85.3 \pm 3.2}$ | $83.4 \pm 5.1$ | $\mathbf{83.4 \pm 5.3}$ | $\mathbf{82.3 \pm 6.7}$ | $77.3 \pm 8.3$ |
| **REDDIT-B** | $79.3 \pm 2.1$ | $\mathbf{80.8 \pm 3.4}$ | $83.9 \pm 1.9$ | $\mathbf{84.1 \pm 2.5}$ | $80.3 \pm 1.6$ | $\mathbf{81.0 \pm 2.8}$ | $81.1 \pm 2.6$ | $\mathbf{81.2 \pm 2.6}$ |

Table 5: Comparison between training on single datasets vs. training on joint datasets. The better result per model is shown in bold face.

The first evaluation aims to assess whether the training of a classifier exclusively on a reference dataset, can lead to a classification accuracy on the aligned datasets superior to that of a random classification (suggesting a possible transfer of knowledge). For each test, we first select the reference dataset and standardize the data with parameters computed on this reference dataset only.

Next, we train the selected model to distinguish classes on the reference dataset and use it in the predictive phase to align the training sets of the other datasets. Once the alignment is complete, we test in the predictive phase the model on both the test set of the reference dataset and the test sets of the aligned datasets. The results of this experiment are visible in Tables 1, 2, 3 and 4 for $k$-NN, RFC, MLP and SVM models, respectively. In these tables, the rows represent the reference datasets used for alignment and training, while the columns represent the datasets on which the learned models are tested.

In the main diagonal of all four tables we notice the highest accuracies. This makes sense as these values represent the models trained and tested on training and test sets stemming from the same dataset. The non-diagonal accuracies reveal the effects of a potential transfer learning. We observe, in general, that the proposed method enables a certain transfer of knowledge. For example, in the first row of Table 1 we see that a $k$-NN trained on DD and then evaluated on the other two datasets leads to a substantial increase of the accuracy when compared to a random classification of 50% (viz. a gain of 18.6 and 26.6 percentage points for MUTAG-M and REDDIT-B, respectively). Considering that this model is not trained directly on these two datasets, this is quite an interesting result. Similar results are observable for other models and datasets (although there are also some cases where obviously no, or only little, transfer of knowledge takes place, e.g. in Table 2 in row 1 where the accuracy remains at 50% for both test sets).

The goal of second experiment is to evaluate whether training the models on a joint dataset can improve the classification performance compared to models trained on single datasets. Hence, after the alignment, we merge the different training sets with the reference dataset, creating a joint dataset for training. Next, we apply standardization on the joint dataset and train the models to distinguish the classes. During the training process, we assign a weight to each dataset, which is 1.0 for data from the reference dataset, and a weight equal to the ratio of the number of elements in the dataset and the total number of elements in the joint dataset for the other datasets. This approach ensures a higher weight to the reference dataset on which we aim to improve the classification performance.

| | MODELS | DD | MUTAG | REDDIT-B |
|---|---|---|---|---|
| GNN | DGCNN (Wang et al., 2019) | $76.8 \pm 3.8$ | $85.4 \pm 2.4$ | $88.2 \pm 2.6$ |
| | DiffPool (Ying et al., 2018) | $75.6 \pm 3.0$ | $82.7 \pm 6.5$ | $89.4 \pm 1.8$ |
| | GIN (Xu et al., 2019) | $76.2 \pm 3.6$ | $\mathbf{86.6 \pm 5.2}$ | $\mathbf{90.3 \pm 1.5}$ |
| | GraphSAGE (Hamilton et al., 2017) | $73.0 \pm 3.0$ | $84.8 \pm 7.3$ | $85.5 \pm 1.5$ |
| NTC | MLP (Errica et al., 2020) | $77.4 \pm 2.2$ | $79.1 \pm 8.1$ | $82.2 \pm 3.0$ |
| | SVM $_{rbf}$ (Gillioz & Riesen, 2023) | $77.3 \pm 3.6$ | $79.6 \pm 7.4$ | $78.3 \pm 2.4$ |
| | $k$-NN $_{L2}$ (Gillioz & Riesen, 2023) | $76.3 \pm 2.7$ | $80.7 \pm 5.9$ | $77.5 \pm 2.9$ |
| | Joint-RFC (Our) | $\mathbf{78.7 \pm 2.7}$ | $85.3 \pm 3.2$ | $84.1 \pm 2.5$ |

Table 6: Average accuracy in a balanced binary classification scenario using the RFC model with joint training compared to state-of-the-art methods viz, *Graph Neural Networks* (GNN) and *Non-Topological Classification* methods (NTC). The best values are highlighted in bold face.

The results of this experiment are shown in Table 5. More comprehensive tables can be found in Appendix A.5. We observe in 11 out of 12 cases a slight increase in performance when the joint datasets is used (rather than the single datasets). The average increase is 0.8 percentage points (ranging from 0.1 percentage points for REDDIT-B and DD using an SVM to 3.7 percentage points for MUTAG-B using an RFC). The observed increase is in line with other studies, e.g., (Verma & Zhang, 2019), in which the methods achieve an increase in performance due to transfer learning between 0.4% and 1.0%.

The somehow limited increase in the classification accuracy could be due to several factors, including the suboptimal alignment of the common space and the relatively small sizes of the training datasets (approximately 1,000, 150, and 1,800 samples for DD, MUTAG, and REDDIT-BINARY, respectively). It is known that larger datasets are often needed to learn and transfer common knowledge. Moreover, in some situations, such as on MUTAG-M trained with the SVM, we even observe a deterioration of the accuracy with the joint dataset. This may be due to the addition of noise along with useful signals when expanding the dataset, which makes it difficult to maintain good performance given the sensitivity of SVMs (Singla & Shukla, 2020). A strategy to mitigate this issue might be to assign higher weights to the reference datasets, or to increase the number of alignment cycles during joint training. For instance, assigning a weight of $10^5$ to the MUTAG-M data during SVM training and after $K = 4$ alignment cycles, the balanced accuracy increases from 77.3% to 80.6% (though with negative effects on the classification accuracy of the aligned datasets).

## 3.3 COMPARISON WITH STATE-OF-THE-ART METHODS

We compare our best model trained on the joint dataset (namely the RFC model) with different state-of-the-art methods, including four graph neural networks, viz. DGCNN (Wang et al., 2019), DiffPool (Ying et al., 2018), GIN (Xu et al., 2019) and GraphSAGE (Hamilton et al., 2017), and three non-topological methods proposed by (Errica et al., 2020) and (Gillioz & Riesen, 2023).

The results in Table 6 show that the RFC model trained on joint datasets shows a performance in line with those obtained with the current state of the art. Specifically, we achieve better results than methods working on non-topological representations (Errica et al., 2020; Gillioz & Riesen, 2023). Moreover, it turns out that our novel method outperforms DiffPool and GraphSAGE on the MUTAG dataset. However, on the REDDIT-B dataset, although it achieves quite good results, our model is still quite far from the best performing GNN models. The results are nevertheless interesting considering that our model works without any topological information.

## 3.4 ABLATION STUDIES

The distribution of the node labels is crucial for our method as it allows the models to effectively transfer the knowledge. For example, we use a mirrored version of MUTAG (MUTAG-M) to better match the distribution of the other two datasets. For our ablation study, we investigate the effect of not using this mirroring of MUTAG.

| | MODELS | DD | MUTAG-M | REDDIT-B |
|---|---|---|---|---|
| Single Dataset | $k$-NN | **71.6 ± 3.2** | 82.0 ± 5.3 | **67.8 ± 1.9** |
| | RFC | **70.3 ± 6.0** | 81.6 ± 5.8 | **67.0 ± 3.9** |
| | MLP | **59.6 ± 8.4** | 83.4 ± 5.1 | **65.8 ± 5.1** |
| | SVM | 50.0 ± 0.0 | 82.3 ± 6.7 | **51.6 ± 3.4** |
| Joint Datasets | $k$-NN | **77.4 ± 2.7** | 82.7 ± 5.3 | **80.0 ± 2.0** |
| | RFC | **78.0 ± 3.3** | **85.3 ± 3.2** | 81.6 ± 3.4 |
| | MLP | **78.3 ± 4.7** | 83.4 ± 5.3 | **78.9 ± 2.5** |
| | SVM | **78.5 ± 2.8** | 77.3 ± 8.3 | **77.2 ± 3.0** |

Table 7: MUTAG mirrored

| | MODELS | DD | MUTAG | REDDIT-B |
|---|---|---|---|---|
| Single Dataset | $k$-NN | 53.9 ± 6.8 | 82.0 ± 5.3 | 51.3 ± 1.9 |
| | RFC | 52.0 ± 5.0 | 81.6 ± 5.8 | 65.0 ± 5.8 |
| | MLP | 49.8 ± 6.8 | 83.4 ± 5.1 | 48.9 ± 4.7 |
| | SVM | 50.0 ± 0.0 | 82.3 ± 6.7 | 50.5 ± 3.2 |
| Joint Datasets | $k$-NN | 76.0 ± 2.4 | 81.0 ± 5.7 | 78.2 ± 2.7 |
| | RFC | 77.9 ± 3.0 | 82.8 ± 3.9 | **83.6 ± 2.5** |
| | MLP | 74.7 ± 4.5 | 80.1 ± 10.3 | 78.2 ± 2.5 |
| | SVM | 72.9 ± 5.0 | **78.0 ± 8.0** | 65.7 ± 4.8 |

Table 8: MUTAG original

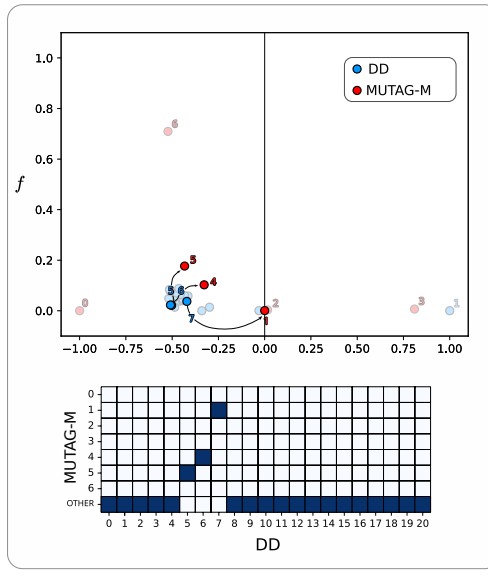
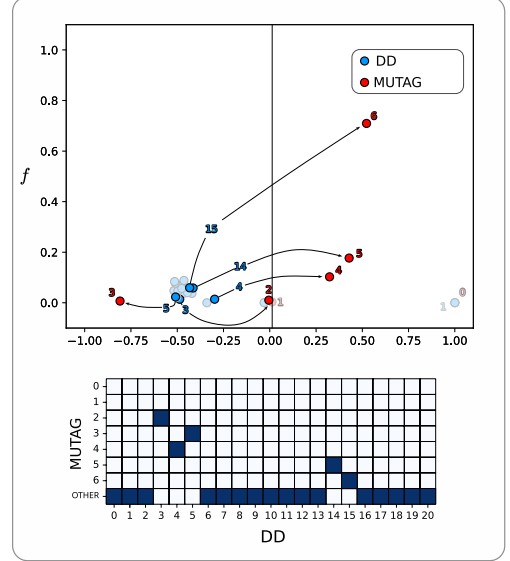

Figure 4: Alignment between MUTAG-M and MUTAG respect DD with $k$-NN.

Tables 7 and 8 show the classification accuracies for both original and mirrored MUTAG, with training on either a single dataset limited to MUTAG (upper part of both tables) or the joint datasets where DD and REDDIT-B are joined to MUTAG (lower part of the tables).

In the upper part of both tables, it can be seen that the mirrored MUTAG (MUTAG-M) provides a more informative alignment, as it allows a better prediction of DD and REDDIT-B than the non-mirrored dataset. This effect is also reflected in the lower part of the tables, where the use of the mirrored MUTAG shows an overall improvement in accuracy in 10 of the 12 possible combinations.

To further confirm that good classification results require a favorable match between the frequency distributions of the labels and their respective class memberships, we analyze the indices obtained from the alignment process. Fig. 4 shows an example of matching DD with MUTAG-M (left side) and DD with MUTAG original (right side) – both using the $k$-NN model. The results of the same analysis with REDDIT-B (rather than DD) are provided in Appendix A.4.

At the top of Fig. 4 we have an illustration that shows the distributions of the two datasets. Moreover, it is depicted which relative frequencies of the reference dataset are matched to which relative frequencies of the aligned dataset. In the lower part of Fig. 4, a matrix displays the relative frequencies of the reference dataset in the rows, while the columns show the relative frequencies of the aligned dataset (both in ascending order). An entry in this matrix is colored if the corresponding frequencies are assigned to each other. For example, it can be seen that the relative frequency of DD with index 7 on the X-axis is assigned to the second relative frequency of MUTAG-M (index 1 on the Y-axis). This information is also shown graphically in the figure above the matrix. Frequencies that are not associated are shown in the last row "OTHER" (made less visible in the plot above the matrix since they are less interesting for our analysis).

In the case of MUTAG-M and DD (on the left), the $k$-NN model does not associate any label of MUTAG-M with the relative frequencies with index 0, 2, 3, and 6 (despite, for instance, the visible proximity of the frequency with index 1 of DD to the frequency with index 3 of MUTAG-M in the upper part of the figure). Yet, on the same dataset DD the indices 5, 6, and 7, show a strong association with MUTAG-M indices. This can also be clearly seen in the upper part of the figure. The index pairs assigned to each other show very similar relative frequencies and a comparable degree of class imbalance in two out of three cases.

Let us now focus on the right side of Fig. 4, which shows the alignment of the DD and MUTAG datasets. We observe that the frequency alignment between the two datasets often involves frequency labels that are quite far away from each other (e.g., index 15 of DD with index 6 of MU-TAG). Clearly, this alignment provides less informative connections, which may explain the poorer performance of MUTAG compared to its mirrored version.

## 4 CONCLUSION AND FUTURE WORK

Both the complexity of the underlying data structures and the scarcity of large datasets pose considerable challenges for the application of transfer learning in the field of graphs. Furthermore, the main limitation for transfer learning with graphs is given by the fact that the nodes in a graph can represent very different entities (e.g., atoms in one graph and users in another graph). This makes it complex to abstract this information and make transfer learning possible.

In this paper, we propose to first represent graphs based on the distribution of their node labels and second align these features using a reference dataset. By means of this specific alignment of the features multi-domain transfer learning can be performed on graphs. In an experimental evaluation, we show that it is possible to transfer at least part of the knowledge between proteins, molecules, and social network datasets. Moreover, we show that joint training on aligned datasets produces a transfer of knowledge that increases the classification performance by an average of 0.8 percentage points. Finally, our evaluation shows that the proposed method is able to outperform methods that rely on non-topological graph representations only.

Although the results are encouraging, the proposed method has some significant limitations. The main limitation is related to the absence of the use of graph topology, which is an important source of information and its non-use might significantly reduce the performances in some datasets. In addition, the high computational complexity of the alignment method hinders its application to more than three datasets. Moreover, the alignment process requires some similarity of distributions and possible mirroring of the underlying classes to fit other datasets. Another important limitation is that the novel representation is not learned directly from a model. This means that it is necessary to use one dataset as a reference to align the others, which can lead to suboptimal results if the reference dataset is not representative enough.

Future research is focused on both overcoming current limitations and finding other parameters that allow information to pass between graphs with labeled nodes from different domains. In particular, we plan to apply the concept discussed in this paper to graph neural networks, thus integrating the topology of graphs. For instance, we could use autoencoders to learn new generic representations of nodes and optimize dataset alignment, reducing dependence on reference datasets and minimizing manual inspections of node label distributions.

## 5 REPRODUCIBILITY STATEMENT

We use $K$-fold cross-validation to evaluate the performance of our models and report the mean and standard deviation. To ensure the repeatability of the experiments, we set the seed of the random generators to 0. The source code, written in Python, is publicly available, commented and accompanied by a documentation that includes instructions for installing dependencies and configuring the virtual environments for running the experiments. The datasets used are public. The hyperparameters used are given both in the source code and in the Appendix( A.2 and A.1 respectively) of this publication.

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

# A APPENDIX

## A.1 CODE

The code used is available at: https://github.com/GraphsAI/tl-graphs-label

## A.2 GRID SEARCH PARAMETERS

Table 9 presents the parameters used to classify the graphs in vector form, selected with attention to the nature of the problem. The vectors represent sparse structures, and the datasets used are of limited size. During joint training, a weight is applied to the data to emphasize the influence of certain samples on model learning. In this context, $k$-NN and RFC prove most suitable, while SVM and MLP are less effective. The choice of parameters for the latter models reflects a delicate balance between avoiding complete overfitting on high-weight data and maintaining generalization capability. With these parameters, we achieve good performance on both joint and individual datasets.

| Model | Tested values | | Alignment Cycles $K$ |
|---|---|---|---|
| $k$-Nearest Neighbors ($k$-NN) | Num. of Neighbors | 5, 7, 9 | 2 |
| | Weights | distance | |
| Random Forest (RFC) | Num. of Estimators | 100, 150, 200 | 2 |
| | Criterion | gini, entropy, log loss | |
| | Min. samples split | 2, 3 | |
| MultiLayer Perceptron (MLP) | Hidden size | 32 | 1 |
| | Learning rate | 0.01 | |
| | Epochs | 50 | |
| Support Vector Machines (SVM) | Kernel | rbf | 1 |
| | C | 10 | |
| | $\gamma$ | (1 / number of features) | |

Table 9: Tested parameters for classifiers

## A.3 DATASETS

Table 10 provides a summary of the characteristics of the datasets used in this study. All datasets are publicly available. For our experiments, we use the datasets available (Ivanov et al., 2019) in the PyTorch-Geometric library (Fey & Lenssen, 2019), a popular research and development framework in the field of deep learning on graphs.

| | Datasets | | |
|---|---|---|---|
| | **DD** | **MUTAG** | **REDDIT-BINARY** |
| Type | Chemical | Chemical | Social networks |
| Num. of Classes | 2 | 2 | 2 |
| Num. of Classes per Node | 89 | 7 | 565 |
| Num. of Graphs | 1178 | 188 | 2000 |
| Avg. Number of Nodes | 284.32 | 17.93 | 429.63 |
| Avg. Number of Edges | 715.66 | 19.79 | 497.75 |

Table 10: Dataset details

### A.4 Alignment Analysis of the Ablation Studies

La Fig. 5 illustrates the index alignment in our ablation studies. The absence of mirroring in the distribution disrupts the uniformity of index alignment, impacting the REDDIT-B dataset as well. This results in less consistent matches, often characterized by distant and less informative associations. Notably, on the left side of the figure, the mirrored MUTAG dataset (MUTAG-M) demonstrates more coherent alignments. In contrast, the right side reveals a negative point from REDDIT-BINARY being matched with a strongly positive component from MUTAG. This misalignment diminishes the informational value, adversely affecting predictive performance on individual datasets and hindering transfer of knowledge between them, ultimately leading to a drop in overall classification accuracy.

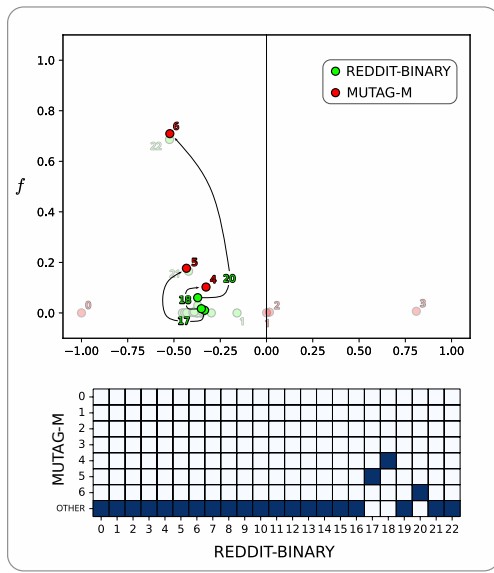 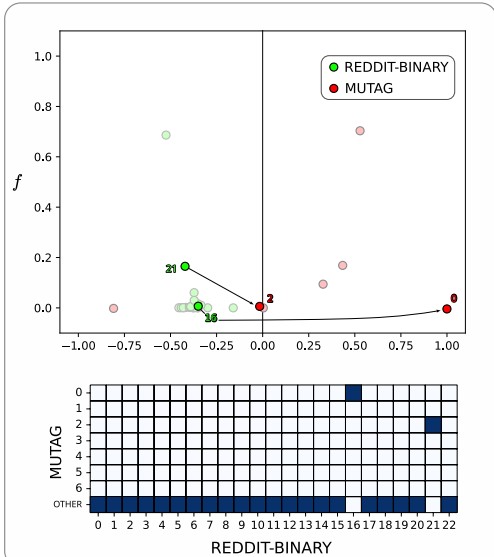

Figure 5: Alignment between DD and REDDIT-BINARY respect MUTAG with $k$-NN.

### A.5 Complete Table of Accuracy on the Joint Dataset

The Tables 11, 12, 13, 14 show the complete results of the joint dataset, providing more detail than the partial analysis in the paper. Columns represent test datasets, while the rows represent reference datasets used to align the others prior to merging. This illustrates how the alignment process influences the performance across the datasets.

| REF. DATASET | | TEST DATASETS | | |
|---|---|---|---|---|
| | | DD | MUTAG-M | REDDIT-B |
| | DD | **78.9 ± 2.1** | 79.8 ± 5.9 | 79.5 ± 2.3 |
| | MUTAG-M | 77.4 ± 2.7 | **82.7 ± 5.3** | 79.9 ± 2.1 |
| | REDDIT-B | 77.8 ± 3.0 | 82.2 ± 5.6 | **80.8 ± 3.4** |

Table 11: $k$-NN model

| REF. DATASET | | TEST DATASETS | | |
|---|---|---|---|---|
| | | DD | MUTAG-M | REDDIT-B |
| | DD | **78.7 ± 2.7** | 82.4 ± 5.1 | 83.6 ± 2.7 |
| | MUTAG-M | 78.0 ± 3.3 | **85.3 ± 3.2** | 81.6 ± 3.4 |
| | REDDIT-B | 78.1 ± 2.3 | 81.2 ± 5.1 | **84.1 ± 2.5** |

Table 12: RFC model

| REF. DATASET | | TEST DATASETS | | |
|---|---|---|---|---|
| | | DD | MUTAG-M | REDDIT-B |
| | DD | **78.0 ± 4.4** | 79.2 ± 10.2 | 80.9 ± 2.4 |
| | MUTAG-M | 78.3 ± 4.7 | **83.4 ± 5.3** | 78.9 ± 2.5 |
| | REDDIT-B | 76.3 ± 3.4 | 80.9 ± 8.3 | **81.0 ± 2.8** |

Table 13: MLP model

| REF. DATASET | | TEST DATASETS | | |
|---|---|---|---|---|
| | | DD | MUTAG-M | REDDIT-B |
| | DD | **80.2 ± 3.3** | **79.33 ± 8.7** | 80.3 ± 2.9 |
| | MUTAG-M | 78.5 ± 2.8 | 77.3 ± 8.3 | 77.2 ± 2.9 |
| | REDDIT-B | 77.9 ± 3.9 | 78.4 ± 9.7 | **81.2 ± 2.6** |

Table 14: SVM model

