# OpenReview forum: "Explainable Transfer Learning on Graphs Using a Novel Label Frequency Representation"
_ICLR.cc/2025/Conference — ICLR 2025 Conference Withdrawn Submission_

### Official Review · Reviewer_Tn3k · 2024-10-28

**Soundness:** 2
**Presentation:** 3
**Contribution:** 3
**Rating:** 5
**Confidence:** 3

**Summary:**

The authors address the challenge of cross-domain transfer learning on graphs and propose a novel method based on label frequency. Specifically, they use node label frequency to create graph-level vectors, which are then used to train additional classifiers, such as kNN, SVM, and MLP. The use of label frequency enhances the method’s explainability. Experimental results demonstrate the method’s superiority over basic graph learning approaches and even GNNs.

**Strengths:**

1. The paper is well-written and easy to follow.

2. The proposed method is novel, addressing an important problem.

3. Experimental results demonstrate the method’s superiority.

**Weaknesses:**

1. The method focuses on frequently occurring patterns across graphs, making it applicable primarily to graph-level tasks. Additionally, it appears to be limited to node-labeled graphs, which restricts its broader applicability.

2. By relying solely on node frequency to define graph properties, the method overlooks the original node features, potentially missing valuable information contained within them.

3. The approach resembles a basic graph learning method using hand-crafted features in a transfer learning setting. The authors should provide additional experimental results to demonstrate the method’s superiority over GNNs in the transfer learning setting.

4. Although the authors aim to address transfer learning across graphs, the experimental results (Tables 1–4) do not show significant performance gains when pretraining on other datasets. Instead, notable negative transfer effects are observed, which may limit the method’s overall contribution.

**Questions:**

1. Can the method be applied to graphs with original node features or to graphs without node labels?

2. Could the authors provide a comparison with GNNs in the transfer learning setting?

---

### Official Review · Reviewer_YpQK · 2024-10-30

**Soundness:** 1
**Presentation:** 2
**Contribution:** 1
**Rating:** 3
**Confidence:** 4

**Summary:**

In this paper, the authors propose a new technique for transfer learning on graphs from different domains. Specifically, the proposed method is based on the relative frequency of node labels. The authors argue that their proposed method can be applied to graphs across domains and is explainable in nature. Experimental results show the effectiveness of the proposed method across domains, such as enhancing social network analysis using chemical and biological graphs.

**Strengths:**

1.	Transfer learning, or more generally, developing universal graph machine learning models across domains is of both theoretical and practical value.
2.	The proposed method is clearly described and easy to understand.

**Weaknesses:**

1.	The technical contribution of the proposed method is severely limited. The proposed relative frequency of node labels is essentially an ad-hoc heuristic method, similar to the term frequency (TF) in natural language processing that dates back to the 20th century. Considering the fast development of graph machine learning and data mining techniques in recent decades, this somewhat antiquated method is of limited novelty and no longer of interest to the general audience.
2.	One particular drawback of the metric is that it does not use any graph structure information, which is vital in graph machine learning. In other words, the proposed method essentially treats a graph as a set of its node labels, disregarding any relational information. Though the authors mention this in the limitation and future direction part, I believe this is a fundamental flaw that is unacceptable for a technical paper for graph machine learning.
3.	Besides, another key drawback of the proposed method is that it is based purely on heuristics and does not have learning ability, which is a major difference between manually designed patterns and machine learning or deep learning based methods.
4.	In experiments, the authors only conduct experiments on TUDataset, which is too small and known to not be able to compare different methods. More experiments on large-scale benchmarks, such as Open Graph Benchmark (OGB), should be adopted to further verify the effectiveness of the proposed method.

All in all, I believe the technical quality of this paper is far from a top-tier conference. If the authors could truly demonstrate the effectiveness of this simple method through comprehensive experiments, it may be possible to rewrite this paper into a “rethinking”-like or “a simple but effective”-like paper, but the current experiments and claims are clearly not sufficient.

**Questions:**

See Weaknesses

---

### Official Review · Reviewer_5gd7 · 2024-11-01

**Soundness:** 2
**Presentation:** 2
**Contribution:** 1
**Rating:** 3
**Confidence:** 3

**Summary:**

The method introduces a new graph representation based on relative node label frequencies for transfer learning. By aligning these vectors in a common space, it enables knowledge transfer between datasets with different graph structures.

**Strengths:**

It is a simple method to recognize the node label frequency as important information for transfer learning tasks. By representing graphs using node label frequency vectors, it maintains simplicity and interpretability while avoiding the need for complex topological information.

**Weaknesses:**

W1. The method ignores node-specific features and structural information, which are critical characteristics of graph data.

W2. Simple counting of node label frequencies may have limitations on large and complex datasets.

W3. Overall, the performance of the proposed method is not consistently competitive with other state-of-the-art approaches.

W4. While the authors claim the method's explainability, it lacks a detailed comparison with post-hoc explanations and interpretable GNNs, and the proposed explanation approach is neither clearly defined nor rigorously evaluated.

**Questions:**

Please refer to W1, W2, W3, and W4.

Q1. What is the unique advantage of the proposed method compared to other transfer learning methods in the graph domain?

Q2. The direction of future work seems orthogonal to the current proposed method. The key idea of the current method is to utilize node label distributions, but this isn't considered in future work. How valuable is the node label distribution overall? And why is the node representation vector considered more promising in the future work?

---

### Official Review · Reviewer_ruqK · 2024-11-04

**Soundness:** 3
**Presentation:** 2
**Contribution:** 2
**Rating:** 3
**Confidence:** 4

**Summary:**

In this paper, the authors propose a universal graph representation learning on relative frequency of the node labels. The representation enables explainable transfer learning between labeled graphs from different domains without the need for additional adaptations.

**Strengths:**

1. The overall presentation is fair and easy to follow the key points in the paper.
2. The proposed model is simple yet interesting to some extend. This may bright some insight for researcher in this field.

**Weaknesses:**

1. This paper aims at explainable transfer learning, however, the entire paper does not discuss what kind of explainability or provide experiments to validate the explainability.
2. The baseline GNN models are relatively old, i.e., the newest one is DGCNN in 2019. More advanced SOTA models in GNN should be compared. More comprehensive experiments are needed to validate the effectiveness of the model.
3. The novelty of this paper is limited, as label frequency strategy is relatively common in the field.

**Questions:**

See weakness above.

---

### Author Response · Authors · 2024-11-19

Dear Reviewers,

I would like to thank you sincerely for the time and effort you have put into evaluating my paper. Your comments were extremely helpful and allowed me to identify significant areas for improvement.

After carefully reflecting on your suggestions and considering the scope of the suggested changes, I believe that a proper review would take longer than expected. Therefore, I have decided to withdraw my paper in order to focus on a more comprehensive review, with the aim of submitting it again in the future, after having thoroughly addressed the critical issues that have emerged.

Thank you again for your feedback and for taking the time to evaluate my work.

Kind regards

---

### Note · Authors · 2024-11-19

I have read and agree with the venue's withdrawal policy on behalf of myself and my co-authors.